

# Interaction and collision of skyrmions in chiral antiferromagnets

George Theodorou[1,2], Bruno Barton-Singer[2] and Stavros Komineas[1,2]

**1** Department of Mathematics and Applied Mathematics,
University of Crete, 70013 Heraklion, Greece
**2** Institute of Applied and Computational Mathematics,
Foundation for Research and Technology - Hellas (FORTH),
Nikolaou Plastira 100, Vassilika Vouton, 70013 Heraklion, Greece

## Abstract

Skyrmions in an antiferromagnet can travel as solitary waves in stark contrast to the situation in ferromagnets. Traveling skyrmion solutions have been found numerically in chiral antiferromagnets. We study head-on collision events between two skyrmions. We find that the result of the collision depends on the initial velocity of the skyrmions. For small velocities, the skyrmions shrink as they approach, then bounce back and eventually acquire almost their initial speed. For larger velocities, the skyrmions approach each other and shrink until they become singular points at some finite separation and are eventually annihilated. Considering skyrmion energetics, we can determine the regimes of the different dynamical behaviors. Using a collective co-ordinate approach, we reproduce the dynamics of the collisions including the variation of the size of the skyrmions and collapse above a critical velocity.



# 1 Introduction

Antiferromagnets (AFM) present magnetic ordering but no net magnetization, in contrast to ferromagnets (FM). As a result, no stray fields are produced and they are robust against moderate external magnetic fields. Their dynamics, such as spin wave frequencies and switching timescales, are in the THz range. These qualities, combined with the fact that antiferromagnetic materials are abundant, are strong motivations for their study. While in the previous decades, antiferromagnetic order had proven difficult to detect and manipulate, interest in antiferromagnetic materials is burgeoning since experimental techniques now allow for optical switching of magnon modes [1–4], optical [5–7] or phonon-based manipulation [8], and electrical [9–11] or current-induced switching of antiferromagnetic domains [11, 12], spin dynamics in the THz range [13–15], and observation of AFM domain walls and their dynamics [16].

The microscopic description of AFM order is formally analogous to the standard description of ferromagnets [17]. Skyrmions and other topological magnetic excitations are expected to form in antiferromagnets in analogy to ferromagnets [18–20]. Specific skyrmionic textures have been observed [21, 22] [23]. AFM skyrmions stand out due to their solitary wave dynamics [24, 25]. The solitary wave character of skyrmions in AFM gives them particle-like properties and they behave as Newtonian or relativistic particles for low and large velocities, respectively. This allows them to be driven to large speeds, e.g., by spin torques [26–28]. In chiral antiferromagnets, they are seen to get elongated as they are accelerated [27, 29, 30], a phenomenon that has been linked to the spiral phase transition [25]. A detailed description of the shape and profile of propagating skyrmions and a full explanation of the phenomenon remain elusive.

In the present paper, we study skyrmion interactions, specifically, the result of a collision between two skyrmions. Collisions of solitons in field theories, mainly applied to particle physics, have revealed surprising behaviors. Two solitons in two or three space dimensions that are colliding are often seen to scatter at right angles [31–34]. This is observed, for example, within the O(3) nonlinear sigma model [35], which is used in particle physics but it is also a simplified model for the antiferromagnetic structure. In the present model, we find that two propagating skyrmions that collide head-on will initially shrink while they are decelerated. At the same time, their configuration changes from elongated to nearly circular. There are two possible outcomes. Slowly moving skyrmions will eventually stop temporarily and bounce back. Skyrmions with larger velocities will shrink completely while approaching each other and they will be annihilated. We discuss the emerging kinetic energy of the spin system that plays a central role in the phenomenon, and we give a formula for its calculation. We employ a collective co-ordinate model that is found to faithfully reproduce the skyrmion collision dynamics including the variation of the skyrmion size during collision and the skyrmion breathing that is observed after the collision. The paper is organized as follows. Section 2 gives a review of the main results for the solitary wave motion of skyrmions. Section 3 contains numerical simulations and a theoretical description of skyrmion collision events. Section 4 give the collective co-ordinate model and applies it for the skyrmion collision. Section 5 contains our concluding remarks. In A, we derive formulae for the energy terms in the continuum model, including a formula for the kinetic energy that is important for the collision dynamics.

# 2 Traveling skyrmions

We study a two-dimensional system and a square lattice of spins $S_{i,j}$ with a fixed length $S_{i,j}^2 = s^2$, where $i, j$ are integer indices for the spin site. A discrete Hamiltonian on the lat-

tice includes symmetric exchange, a Dzyaloshinskii-Moriya (DM), and an easy-axis anisotropy term, $E^d = E^d_{\text{ex}} + E^d_{\text{DM}} + E^d_{\text{a}}$, with

$$
\begin{aligned}
E^d_{\text{ex}} &= J \sum_{i,j} \boldsymbol{S}_{i,j} \cdot (\boldsymbol{S}_{i+1,j} + \boldsymbol{S}_{i,j+1}), \\
E^d_{\text{DM}} &= D \sum_{i,j} \left[ \hat{\boldsymbol{e}}_2 \cdot (\boldsymbol{S}_{i,j} \times \boldsymbol{S}_{i+1,j}) - \hat{\boldsymbol{e}}_1 \cdot (\boldsymbol{S}_{i,j} \times \boldsymbol{S}_{i,j+1}) \right], \\
E^d_{\text{a}} &= -\frac{K}{2} \sum_{i,j} [(\boldsymbol{S}_{i,j})_z]^2,
\end{aligned}
\tag{1}
$$

where $\hat{\boldsymbol{e}}_\mu$, $\mu = 1, 2, 3$ denote the unit vectors in spin space, $(\boldsymbol{S}_{i,j})_z$ is the out-of-plane component of a spin vector, and $J, D, K$ are positive constants. The equations of motion for each spin are derived from the energy and have the form

$$
\frac{\partial \boldsymbol{S}_{i,j}}{\partial t} = -\boldsymbol{S}_{i,j} \times \frac{\partial E}{\partial \boldsymbol{S}_{i,j}},
\tag{2}
$$

where the effective field is

$$
\begin{aligned}
-\frac{\partial E^d}{\partial \boldsymbol{S}_{i,j}} = &-J(\boldsymbol{S}_{i+1,j} + \boldsymbol{S}_{i,j+1} + \boldsymbol{S}_{i-1,j} + \boldsymbol{S}_{i,j-1}) \\
&+ D \left[ \hat{\boldsymbol{e}}_2 \times (\boldsymbol{S}_{i+1,j} - \boldsymbol{S}_{i-1,j}) - \hat{\boldsymbol{e}}_1 \times (\boldsymbol{S}_{i,j+1} - \boldsymbol{S}_{i,j-1}) \right] + K(\boldsymbol{S}_{i,j})_3 \hat{\boldsymbol{e}}_3.
\end{aligned}
\tag{3}
$$

A continuum model for the antiferromagnet will facilitate the analysis and it will offer significant insight into the dynamics. We consider a tetramerization of the square lattice and define the four linear combinations of the spins at each tetramer [19, 25],

$$
\begin{aligned}
\boldsymbol{m} &= \frac{1}{4s}(\boldsymbol{S}_{i,j} + \boldsymbol{S}_{i+1,j} + \boldsymbol{S}_{i+1,j+1} + \boldsymbol{S}_{i,j+1}), & \boldsymbol{n} &= \frac{1}{4s}(\boldsymbol{S}_{i,j} - \boldsymbol{S}_{i+1,j} + \boldsymbol{S}_{i+1,j+1} - \boldsymbol{S}_{i,j+1}), \\
\boldsymbol{k} &= \frac{1}{4s}(\boldsymbol{S}_{i,j} + \boldsymbol{S}_{i+1,j} - \boldsymbol{S}_{i+1,j+1} - \boldsymbol{S}_{i,j+1}), & \boldsymbol{l} &= \frac{1}{4s}(\boldsymbol{S}_{i,j} - \boldsymbol{S}_{i+1,j} - \boldsymbol{S}_{i+1,j+1} + \boldsymbol{S}_{i,j+1}).
\end{aligned}
\tag{4}
$$

A small parameter $\epsilon$ is introduced that will control the continuum approximation. The scaled space and time variables are defined as

$$
x = \epsilon i, \qquad y = \epsilon j, \qquad \tau = 2\sqrt{2}\epsilon s J t.
\tag{5}
$$

The continuum model is written entirely in terms of the Néel vector $\boldsymbol{n} = \boldsymbol{n}(x, y, \tau)$ with components $(n_1, n_2, n_3)$, which satisfies the local constraint $|\boldsymbol{n}| = 1$. This obeys an extension of the $\sigma$ model [17, 19],

$$
\boldsymbol{n} \times (\ddot{\boldsymbol{n}} - \boldsymbol{f}) = 0, \qquad \boldsymbol{f} = \Delta \boldsymbol{n} + 2\lambda \epsilon_{\mu\nu} \hat{\boldsymbol{e}}_\mu \times \partial_\nu \boldsymbol{n} + n_3 \hat{\boldsymbol{e}}_3, \qquad \lambda = \frac{D}{\sqrt{KJ}},
\tag{6}
$$

where the dot denotes differentiation with respect to the scaled time $\tau$, $\Delta$ denotes the Laplacian in two dimensions, $\epsilon_{\mu\nu}$ is the antisymmetric tensor with $\mu, \nu = 1, 2$, and the summation convention for repeated indices is adopted. In deriving model (6), we have made the choice

$$
\epsilon = \sqrt{\frac{K}{J}}.
\tag{7}
$$

The fields $\boldsymbol{m}, \boldsymbol{k}, \boldsymbol{l}$ are auxiliary and they are given in terms of $\boldsymbol{n}$ as

$$
\boldsymbol{m} = \frac{\epsilon}{2\sqrt{2}} \boldsymbol{n} \times \dot{\boldsymbol{n}}, \qquad \boldsymbol{k} = -\frac{\epsilon}{2} \partial_1 \boldsymbol{n}, \qquad \boldsymbol{l} = -\frac{\epsilon}{2} \partial_2 \boldsymbol{n}.
\tag{8}
$$

The system is Hamiltonian with energy $E = E_{\text{kin}} + V$, where the kinetic and potential energies are

$$E_{\text{kin}} = \frac{1}{2} \int \dot{\boldsymbol{n}}^2 \, d^2 x \,, \qquad V = E_{\text{ex}} + E_{\text{DM}} + E_a \,, \tag{9}$$

$$E_{\text{ex}} = \frac{1}{2} \int (\partial_\mu \boldsymbol{n}) \cdot (\partial_\mu \boldsymbol{n}) \, d^2 x \,, \quad E_{\text{DM}} = -\lambda \int \epsilon_{\mu\nu} \hat{\boldsymbol{e}}_\mu \cdot (\partial_\nu \boldsymbol{n} \times \boldsymbol{n}) \, d^2 x \,, \quad E_a = \frac{1}{2} \int (1 - n_3^3) \, d^2 x \,.$$

The effective field in Eq. (6) is derived from $\boldsymbol{f} = -\delta V / \delta \boldsymbol{n}$.

The static sector of the $\sigma$ model (6) for the Néel vector is identical to the static sector of the Landau-Lifshitz equation for the magnetization vector of a ferromagnet with corresponding interactions. Therefore, static skyrmions are obtained in an AFM for parameter values $\lambda < 2/\pi$. For $\lambda > 2/\pi$, a spiral is the ground state.

Skyrmions traveling with velocity $v$ are dynamical solutions of Eq. (6) of the form $\mathbf{n}(x - v\tau, y)$, where we have chosen $x$ as the direction of propagation. We denote such solutions

$$\mathbf{n}_v(\xi, \eta) \,, \qquad \xi = x - v\tau \,, \quad \eta = y \,. \tag{10}$$

The skyrmion velocity can take values in the range $0 \le v < v_c \equiv \sqrt{1 - (\pi\lambda/2)^2}$ [25].

We have simulated numerically the solitary wave motion of a skyrmion. We consider a solution (10) of the continuum model (6) as obtained in [25]. The auxiliary fields are given by (8), e.g., we have $\boldsymbol{m} = -\frac{\epsilon v}{2\sqrt{2}} \boldsymbol{n} \times \partial_1 \boldsymbol{n}$ with $\boldsymbol{n} = \boldsymbol{n}_v$. We use finite differences and apply the configuration on a lattice for the variables (4). The corresponding spin variables are obtained by inverting Eq. (4) on a numerical mesh. The simulations are performed using Eqs. (2) for the spins. We typically use the parameter values

$$J = 1 \,, \qquad D = 0.015 \,, \qquad K = 0.0009 \,. \tag{11}$$

Then, Eq. (7) gives $\epsilon = 0.03$ and Eq. (6) gives $\lambda = 0.50$. If we have a velocity $v$ in the scaled variables used in the continuum, then we expect a skyrmion velocity $v^d$ in the simulation lattice, given via

$$v = \frac{\Delta x}{\Delta \tau} = \frac{\epsilon \Delta i}{2\sqrt{2} \epsilon \Delta t} = \frac{v^d}{2\sqrt{2}} \,, \qquad v^d = \frac{\Delta i}{\Delta t} \,. \tag{12}$$

Numerical simulations verify that a traveling skyrmion, as calculated in the continuum model, is propagating coherently with the expected velocity in the spin lattice. We measure an error in the expected velocity approximately 2% for initial velocity $v = 0.2$ and 1% for initial velocity $v = 0.4$.

The energy terms play an important role in the interpretation of the simulation results. For this purpose, we need to obtain the relation between the discrete, Eq. (1), and the continuum, Eq. (9), energy terms. We also need formulae to approximate the continuum energy terms on the numerical mesh. These are derived in A. The main new element is formula (A.4) for the kinetic energy calculated in terms of spin variables. Fig. 1 gives the energy of propagating skyrmions versus their velocity for $\lambda = 0.50$, calculated numerically. The value $4\pi$ is the minimum exchange energy possible. A static skyrmion has total energy $E < 4\pi$ due to the negative contribution of the DM term. In the limit of small radius (obtained for small $\lambda$), the energy of a static skyrmion approaches the value $E = 4\pi$ [36, 37]. The energy of a traveling skyrmion becomes $E > 4\pi$ for a large enough velocity, for example, for $v > 0.53$ when $\lambda = 0.50$. This is a critical velocity value because skyrmions with energy greater than $4\pi$ can potentially shrink to become singular configurations and thus physically disappear [38].

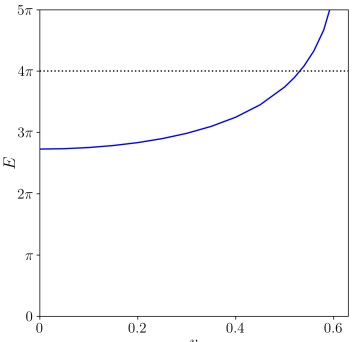

Figure 1: Energy $E$ of propagating skyrmions of the form (10) as a function of velocity $v$ for parameter value $\lambda = 0.50$. The static skyrmion ($v = 0$) has energy $E < 4\pi$. The energy crosses the critical value $E = 4\pi$ at $v \approx 0.53$.

## 3 Collisions of skyrmions

Propagating skyrmions in antiferromagnets can be obtained through acceleration induced by an energy gradient. This can be produced by, e.g., an anisotropy gradient [39] or any other variation of the material properties. One could consider a voltage that would change the anisotropy or DM strength locally. If two (or more) skyrmions are at rest, a local change in the magnetic energy would cause an acceleration of both skyrmions towards the lowest energy region. If this region is between them, that would automatically set the skyrmions on a head-on collision course. In contrast to the above, collisions of skyrmions in ferromagnets cannot occur in the standard Newtonian way even if they are manipulated [40].

We will set up our system assuming that the two skyrmions are moving toward each other with an initial velocity and they are in a homogeneous background (that is, any voltage that was initially applied in order to obtain acceleration has been switched off). We initialize the numerical simulations using the Ansatz

$$n(x, y, 0) = \begin{cases} n_v(x - X_0/2, y), & x \geq 0, \\ n_v(x + X_0/2, y), & x < 0, \end{cases} \tag{13}$$

where $n_v$ is the traveling skyrmion solution in Eq. (10) and $X_0$ is the initial distance between the skyrmions. Ansatz (13) represents two skyrmions at positions $(\pm X_0/2, 0)$ at $\tau = 0$, each one has a speed $v$ and they are on a collision course. We typically choose $X_0 = 21$ in the simulations. The auxiliary fields are given by Eq. (8). Specifically, the magnetization is

$$m(x, y, 0) = \begin{cases} \frac{\epsilon v}{2\sqrt{2}} n \times \partial_1 n, & x \geq 0, \\ -\frac{\epsilon v}{2\sqrt{2}} n \times \partial_1 n, & x < 0. \end{cases} \tag{14}$$

The corresponding spin configuration is reconstructed by inverting Eqs. (4), and the simulation is performed on the spin lattice. For the time evolution, we use Eqs. (2). Our typical numerical mesh is $2100 \times 700$. Since, $\epsilon = 0.03$, the lattice extends in space in $-31.5 < x < 31.5$, $-10.5 < y < 10.5$. Simulations performed on a larger lattice did not show any change in the results.

Figure 2 shows snapshots of the Néel vector configuration for a head-on collision of skyrmions with initial velocities $v = 0.2$ [41]. Only the central part of the numerical mesh is shown in the snapshots. The skyrmions are initially at positions $(\pm 10.5, 0)$. The first snapshot

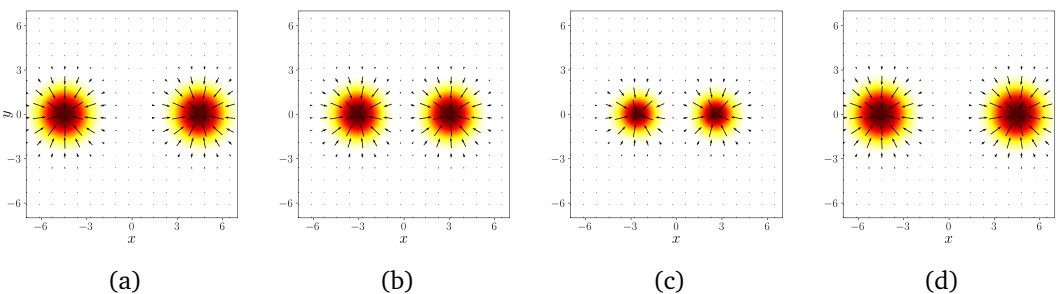

(a)  (b)  (c)  (d)

Figure 2: Simulation of a head-on collision of skyrmions place initially ($\tau = 0$) at positions $(\pm 10.5, 0)$ with velocities $v = \mp 0.2$. (a) At the time $\tau = 30.0$, the skyrmions are at a distance, not yet interacting. (b) At $\tau = 38.0$, their speed and size have decreased due to interaction. (c) At $\tau = \tau_{\text{crit}} = 46.2$, they stop ($v = 0$) temporarily and have a minimum size. (d) At $\tau = 60.8$, they have bounced back and they have acquired almost their initial speed and size.

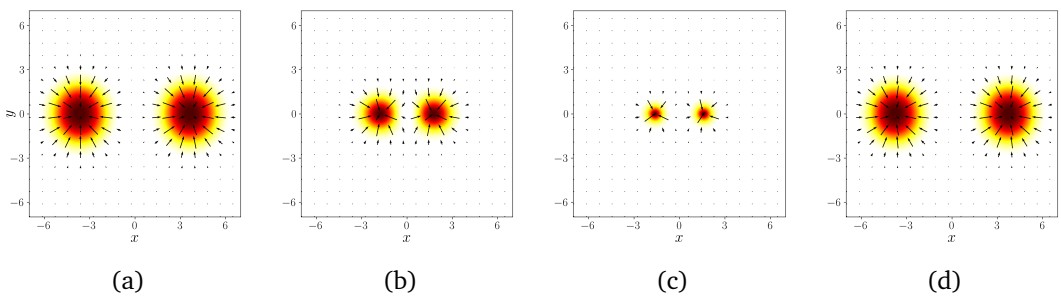

(a)  (b)  (c)  (d)

Figure 3: Simulation of a head-on collision of skyrmions placed initially ($\tau = 0$) at positions $(\pm 10.5, 0)$ with velocities $v = \mp 0.4$. The process is similar to Fig. 2. We show snapshots at times (a) $\tau = 17.1$, (b) $\tau = 23.7$, (c) $\tau = \tau_{\text{crit}} = 27.1$, where the skyrmions have approached closer and their size is smaller compared to the corresponding picture in Fig. 2, and at (d) $\tau = 37.2$.

shows the skyrmions when they are far enough from each other and they are not yet interacting. The second snapshot shows that, after approaching each other, the size of both skyrmions has been reduced due to the interaction. Their velocity is also reduced. The third snapshot shows the skyrmions at the minimum distance when their velocity is almost zero, while their size has further been reduced. After this moment, the skyrmions bounce back, their speed increases and their size grows again. The last snapshot shows the skyrmions when they are at a large distance from each other and they have been restored to almost their original profile, size, and speed.

Figure 3 presents the results of a second simulation for initial velocity $v = 0.4$ [41]. The simulation results are similar to the case of $v = 0.2$ in Fig. 2. But, the skyrmions, at the distance of minimum separation, shown in the third snapshot, have reached a minimum size that is now smaller. The phenomenon of bouncing back of skyrmions after a collision has been noted in [32] in a $\sigma$ model with Skyrme-like terms.

Figure 4 shows the positions of the skyrmions on the $x$ axis with respect to time. For the skyrmion position, we find the two points on the $x$-axis where $n_3 = 0$ and take the middle point between the two. The slope of the curves shows that the skyrmions acquire almost their initial velocity after bouncing back. For initial velocity $v = 0.2$, they eventually get to a velocity $v = 0.192$ after bouncing back, and for initial velocity $v = 0.4$, they get to $v = 0.384$.

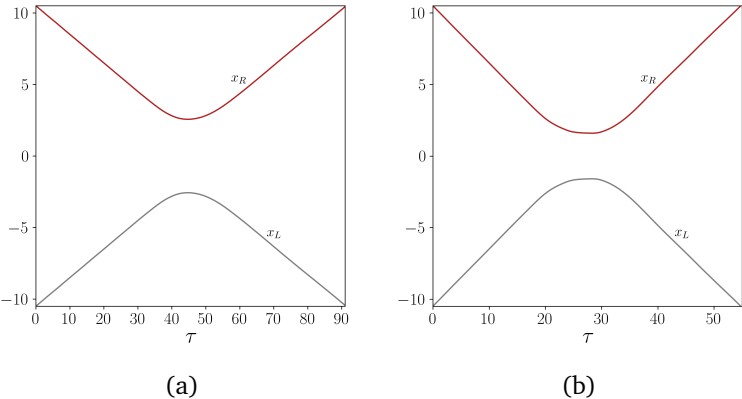

(a)           (b)

Figure 4: The position of the two colliding skyrmions on the $x$ axis ($x_L, x_R$ for the skyrmions coming from the left and right, respectively) as a function of time for initial velocities (a) $v = 0.2$, corresponding to the simulation in Fig. 2, and (b) $v = 0.4$, corresponding to the simulation in Fig. 3. The skyrmions stop temporarily and bounce back after the collision.

Figure 5 shows the radius of the skyrmions during the collision. The radius $R$ is calculated as half the distance between the two points on the $x$ axis where $n_3 = 0$, for each skyrmion. The skyrmion radius is almost constant before the collision, it reduces significantly during the collision and it is later restored to its initial size. After the collision, the skyrmion radius oscillates, clearly indicating that the breathing mode [38,42] has been excited. The numerical simulations show that the decrease in the skyrmion size due to the interaction is a generic phenomenon. The small oscillations are mainly attributed to the error introduced by the transformation of variables from the continuum model, where the solitary wave solutions have been found, to the spin lattice, where the simulations are performed.

A qualitative description of the skyrmion dynamics can be given by following the evolution of the energy terms. When the skyrmions approach, they decelerate, and their kinetic energy is converted to potential energy. This can be considered to consist of internal and interaction energy. The internal energy of the skyrmions is increasing as the skyrmion size is decreasing from its equilibrium value. Fig. 6 shows the evolution of the energy terms for the simulation in Fig. 2 with initial skyrmion velocity $v = 0.2$. The DM and anisotropy energies are decreasing in absolute value as the size of the skyrmions decreases during the collision, while the exchange energy approaches its minimum possible value which is $8\pi$. The kinetic energy is shown in Fig. 6b and it approaches zero (to within numerical accuracy) at the time of minimum distance between the skyrmions.

The calculation of the energy terms is based on the formulae for the potential energy terms on the spin lattice (1). The kinetic energy, which is only meaningful in the continuum theory, has to be defined in a consistent way on the spin lattice. This is accomplished in A and the formula for the kinetic energy $E_{\text{kin}}$ calculated via the spin vectors is given in Eq. (A.4). The exchange energy plotted in Fig. 6 is $E_{\text{ex}} = E_{\text{ex}}^d - E_{\text{kin}}$, while the DM and anisotropy terms are calculated using the formulae in Eq. (1), that is, we use the approximations $E_{\text{DM}} = E_{\text{DM}}^d$, $E_{\text{a}} = E_{\text{a}}^d$.

We have performed simulations of head-on collisions also with higher initial skyrmion velocities. For example, we choose an initial velocity $v = 0.56$ [41]. From Fig. 1, we see that such propagating skyrmions have energy $E > 4\pi$ each, which is a crucial energy bound. Simulations show that such fast-moving skyrmions approach each other and their size decreases until they become very small with a size comparable to the lattice spacing. The skyrmions are then annihilated and the energy is eventually converted to spin waves. The collapse of solitons

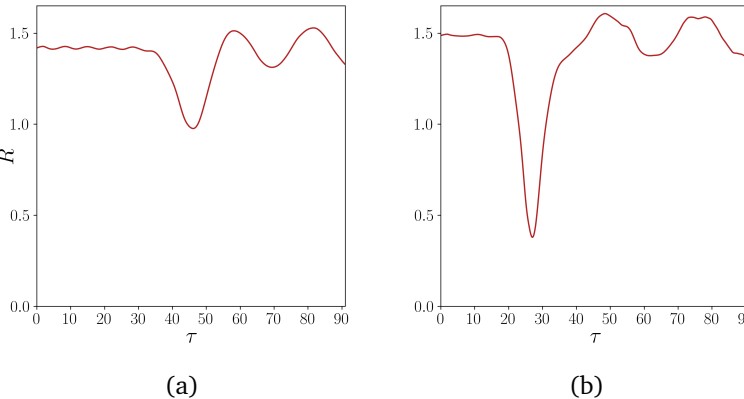

Figure 5: The radius $R$ of the skyrmions during the collision for the cases with initial velocities (a) $v = 0.2$, shown in Fig. 2, (b) $v = 0.4$, shown in Fig. 3. The skyrmion radius is reduced during collision. The breathing mode has been excited after the skymions bounce back.

has been studied rigorously within certain models in [43, 44], while a skyrmion annihilation phenomenon within the present model was discussed in [38].

The process can be understood based on the conversion of kinetic energy to exchange energy. Since each skyrmion has a total energy $E > 4\pi$, the conversion of kinetic to potential energy is continuing during collision until the skyrmions have an infinitesimally small size with exchange energy $E_{ex} = 4\pi$ and $E_{DM} = 0, E_a = 0$. This is observed to happen in all simulations with large skyrmion initial velocities for which the energy of each of the initial propagating skyrmions is $E > 4\pi$. Different outcomes of the simulations, such as a combination of propagation and breathing of skyrmions cannot easily be excluded by theoretical arguments, however, no outcome other than the eventual skyrmion annihilation has been observed in our simulations.

The generic phenomenon of skyrmion shrinking during collision will be addressed in the next section using a collective co-ordinate method. On a descriptive level, we note that the interaction is due to the overlap of the far field of one skyrmion with the core of the other one. Eventually, the skyrmions decelerate and shrink. On a phenomenological level, this seems similar to the squeezing of two elastic balls during their collision, except that the skyrmions interact already before their cores touch each other.

## 4 Skyrmion elasticity and dynamics

### 4.1 Collective co-ordinates

We will show that the response of the skyrmions during collision is driven by their elasticity property which has been previously manifested as a breathing oscillation mode [38, 42]. This is analogous to kink-antikink collisions, where the kink's breathing oscillation mode has proved crucial to understanding the emergent dynamics [45]. We describe the two-skyrmion configuration during the collision process by a collective co-ordinate ansatz [46]

$$\boldsymbol{n}(\boldsymbol{x}, t) = \boldsymbol{n}\left(X(t), R(t)\right), \tag{15}$$

where $\boldsymbol{n}(X, R)$ is a field configuration of two skyrmions at distance $X$ from each other, both with radius $R$. We approximate this configuration by single skyrmion profiles which are placed

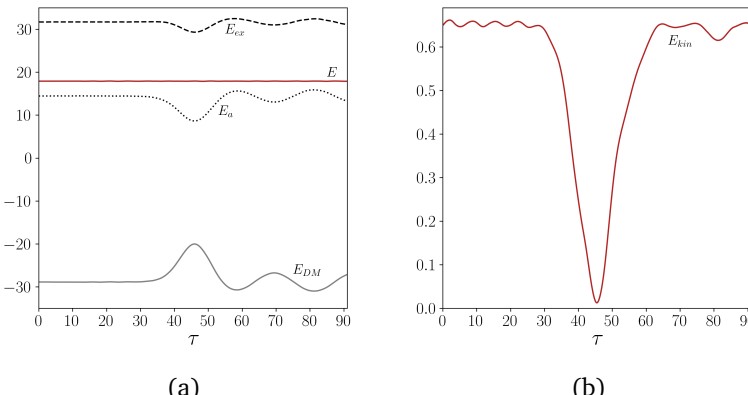

Figure 6: The evolution of the energy terms for the simulation of the collision of skyrmions with initial velocity $v = 0.2$. (a) The total energy $E$ is conserved during the collision. Also plotted are the exchange, DM, anisotropy energy, and (b) the kinetic energy.

in the left and in the right half plane,

$$n(X,R) = \begin{cases} n_R(x + \frac{X}{2}, y), & x < 0, \\ n_R(x - \frac{X}{2}, y), & x > 0. \end{cases} \tag{16}$$

We confine the ansatz to skyrmion configurations $n_R = n(r; \lambda)$ that are obtained as axisymmetric static solutions of the model for a parameter $\lambda$ that gives the required radius $R$. For every axisymmetric skyrmion profile the radius $R(\lambda)$ is defined to be at the point where the third magnetization component is $n_3(r = R; \lambda) = 0$. The radius $R$ for the static solutions $n_R$ is linked with the DM strength $\lambda$ via a monotonic but nontrivial relation [47]. Ansatz (16) is expected to be a good approximation when (i) the skyrmions are well-separated, i.e., $X \gg R$, so that their interaction is weak, and (ii) each skyrmion velocity is small, so that the profiles of the moving skyrmions are close to the profiles of the static ones [33,48].

We will study the dynamics of the two collective co-ordinates $X, R$. We start with the Lagrangian of the theory

$$L = E_{\text{kin}} - V.$$

Substituting the ansatz (16), the Lagrangian becomes a function of $X, R$.

## 4.2 Potential energy

For well-separated skyrmions, $X \gg R$, we write the total potential energy for the skyrmion pair as a sum of an elastic and an interaction part,

$$V(X,R) = 2V_{\text{el}}(R) + V_{\text{int}}(X,R). \tag{17}$$

The elastic part $V_{\text{el}}$ refers to the potential energy (9) of an isolated skyrmion and it depends only on the radius $R$. We choose a parameter value $\lambda_0$ for the model and denote the equilibrium radius of the static skyrmion by $R_0$. The elastic energy has a parabolic form when the skyrmion has a radius $R$ that is close to its equilibrium value $R_0$,

$$V_{\text{el}}(R) \approx V_{\text{el}}(R_0) + \frac{1}{2} \partial_R^2 V_{\text{el}}|_{R=R_0} (R - R_0)^2. \tag{18}$$

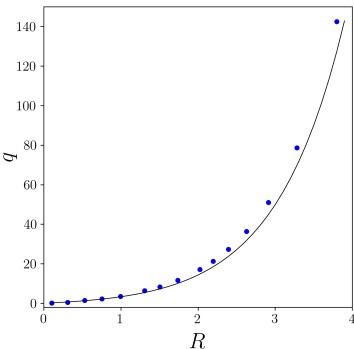

Figure 7: The factor $q$ of the skyrmion far field in Eq. (19) as a function of the skyrmion radius $R$. Numerical results are shown by circles. The solid line shows the function $q(R) = 2/K_1(R)$ which fits the numerical results very well for all $R$.

Let us now describe our method for calculating $\partial_R^2 V_{\text{el}}$ and other quantities of interest. Firstly, we find numerically the profiles of the static skyrmions $\boldsymbol{n}_R = \boldsymbol{n}(r; \lambda)$ for all $\lambda$ by the shooting method. We can then calculate $V_{\text{el}}$ for a range of radii around $R_0$. Finally, we fit the resulting potential curve by a polynomial and find the quadratic factor which directly gives $\partial_R^2 V_{\text{el}}|_{R_0}$.

The interaction part $V_{\text{int}}$ is due to the overlap between the two skyrmions. When they are well-separated, it depends on the far field of a skyrmion, which is conveniently given in terms of the polar angle $\Theta(r)$ in spherical co-ordinates of the vector $\boldsymbol{n}$ as (see, e.g., [37,47])

$$\Theta(r) = q(R) K_1(r). \tag{19}$$

The interaction energy can then be approximated by a standard method [49–51]:

$$V_{\text{int}}(X, R) = 2\pi q(R)^2 K_0(X) + O\left(e^{-\frac{3}{2}X}\right). \tag{20}$$

The correction term has unknown dependence on $R$. In the cited works, the interaction derived in this way had good agreement with direct calculation down to $X \sim 2$.

We find $q(R)$ by fitting the numerical profiles at large $r$ to the formula in Eq. (19). In Fig. 7, we plot $q$ as a function of the equilibrium radius $R$. In the same figure, the formula $q(R) = 2/K_1(R)$ is plotted. We expect $q = 2R$ for small radius, and $q = e^R \sqrt{8R/\pi}$ for large radius [52]. We see that this formula not only gives the correct behavior at small and large $R$ but is also an excellent fit for all intermediate $R$.

### 4.3 Kinetic energy

For the kinetic energy, we insert the ansatz (15) in Eq. (9) and obtain

$$E_{\text{kin}} = \frac{1}{2} g_{XX} \dot{X}^2 + \frac{1}{2} g_{RR} \dot{R}^2 + g_{RX} \dot{R} \dot{X}, \tag{21}$$

where the elements of the metric are

$$g_{XX} = \int |\partial_X \boldsymbol{n}|^2 d^2 x, \qquad g_{RR} = \int |\partial_R \boldsymbol{n}|^2 d^2 x, \qquad g_{RX} = \int \partial_R \boldsymbol{n} \cdot \partial_X \boldsymbol{n} \, d^2 x. \tag{22}$$

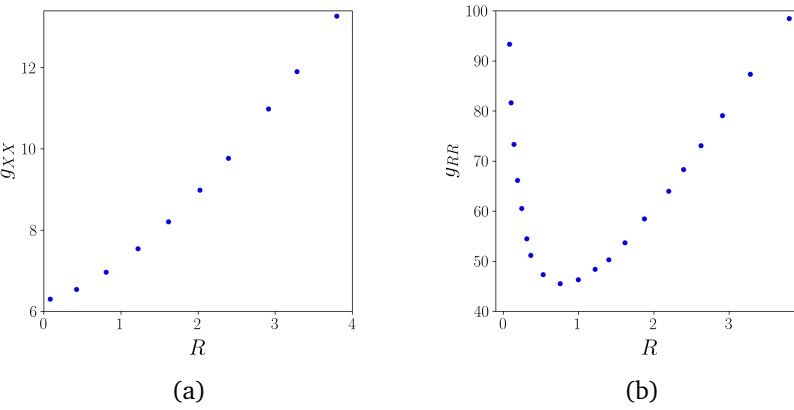

Figure 8: (a) The metric element $g_{XX}$ (related to the mass of skyrmions) as a function of $R$ found numerically. It approaches $2\pi$ for $R \to 0$ and it grows linearly $g_{XX}(R) = \pi R$ for large $R$. (b) The metric element $g_{RR}$ as a function of $R$ found numerically. For small radius, we expect it to diverge as $g_{RR} = -8\pi \ln R$ and it grows linearly $g_{RR} = 8\pi R$ for large $R$. A minimum is reached at $R \approx 0.75$.

For calculating $g_{XX}$, we observe that, by our construction in Eq. (16), $\partial_X \boldsymbol{n} = +\frac{1}{2}\partial_1 \boldsymbol{n}$ in the left half and $-\frac{1}{2}\partial_1 \boldsymbol{n}$ in the right half of the plane. The integration should be done on the respective half plane for each skyrmion $\boldsymbol{n}_R$. This can be extended to the whole plane with a small error, and we obtain

$$g_{XX} = \frac{1}{4}\int |\partial_1 \boldsymbol{n}|^2 d^2x \simeq \frac{1}{2}\int |\partial_1 \boldsymbol{n}_R|^2 d^2x\,.$$

Since the single-skyrmion configuration is axially symmetric, it follows that $|\partial_1 \boldsymbol{n}_R| = |\partial_2 \boldsymbol{n}_R|$, and thus

$$g_{XX} = \frac{1}{2}E_{\text{ex}}(\boldsymbol{n}_R)\,. \tag{23}$$

The result depends on the single-skyrmion configuration and thus $g_{XX} = g_{XX}(R)$. Fig. 8a shows the numerically calculated $g_{XX}$ as a function of the skyrmion radius $R$. The quantity $g_{XX}$ is usually interpreted as being proportional to the mass of the skyrmion [31]. Using formulae from asymptotic analysis for $E_{\text{ex}}$, we expect $g_{XX} \to 2\pi$ for small radius and linear growth, $g_{XX} = \pi R$, for large radius [37,47], in agreement with the results shown in Fig. 8a.

For $g_{RR}$, we find $\partial_R \boldsymbol{n}$ using the numerically calculated profiles $\boldsymbol{n}_R$ and applying finite differences. In the approximation that the configuration of each skyrmion extends in the entire plane, we have

$$g_{RR} = 2\int |\partial_R \boldsymbol{n}_R|^2 d^2x\,,$$

and it is thus a function of $R$ only. The result is shown in Fig. 8b. For large $R$, the skyrmion can be approximated as a circular domain wall at distance $R$ from the skyrmion center so that $\partial_R \boldsymbol{n}_R \approx -\partial_r \boldsymbol{n}_R$ and $g_{RR} \approx 2\int |\partial_r \boldsymbol{n}_R|^2 d^2x = 4E_{\text{ex}}(\boldsymbol{n}_R)$. Using asymptotic results for the energy [52], we have $g_{RR} = 8\pi R$ for $R \gg 1$, and it can be shown that $g_{RR} = -8\pi \ln R$ for $R \gg 1$ [38]. The asymptotic results indicate that $g_{RR}$ should have a minimum at an intermediate radius, which is numerically found to be at $r \approx 0.75$.

By using the rotation symmetry of the skyrmion, $\boldsymbol{n}_R(\boldsymbol{x}) \mapsto -\boldsymbol{n}_R(-\boldsymbol{x})$, we have

$$g_{XR} = 0\,.$$

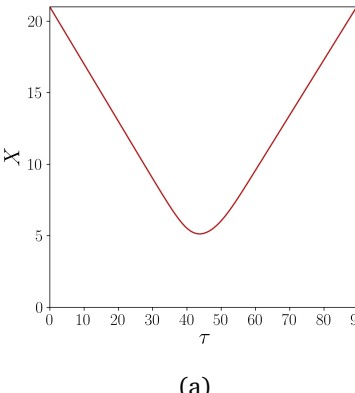
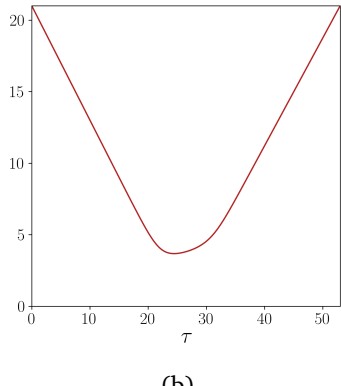

(a)                          (b)

Figure 9: The distance between the two colliding skyrmions as a function of time according to the collective co-ordinate model for initial velocities (a) $v = 0.2$ and (b) $v = 0.4$. The same initial conditions are used as in Fig. 4. (For comparison with Fig. 4, it is $x_L = -X/2$, $x_R = X/2$ for the skyrmions coming from the left and right, respectively.)

Collecting the results for the metric elements and the potential energy, we have the Lagrangian

$$L(X, R, \dot{X}, \dot{R}) = \frac{1}{2} g_{XX}(R)\dot{X}^2 + \frac{1}{2} g_{RR}(R)\dot{R}^2 - 2V_{\text{el}}(R) - V_{\text{int}}(X, R). \tag{24}$$

The dynamics are given by the system of Euler-Lagrange equations

$$g_{XX}(R)\ddot{X} = -\frac{\partial V_{\text{int}}}{\partial X}(X, R), \tag{25a}$$

$$g_{RR}(R)\ddot{R} + \frac{1}{2} g_{RR}'(R)\dot{R}^2 = -2V_{\text{el}}'(R) - \frac{\partial V_{\text{int}}}{\partial R}(X, R). \tag{25b}$$

In this model, we effectively assume that, at each moment in time, the profile of a skyrmion is that of a static solution $\boldsymbol{n}_R$, obtained in a model for some $\lambda$ that corresponds to the skyrmion instantaneous radius $R = R(\lambda)$.

Since $q(R)$ is a monotonically increasing function, the influence of one skyrmion on the other will create a force causing the skyrmion radii to decrease, which will grow stronger as the skyrmions get closer. If the skyrmions rebound, this coupling will excite the breathing mode of the skyrmion. Otherwise, this model allows for the radius to reach zero in finite time, meaning the skyrmions collapse. Therefore, this collective co-ordinate model contains a qualitative explanation of the observed numerical behaviour.

## 4.4 Solution of the collective co-ordinate model

We now investigate numerically the dynamics of a skyrmion as given by Eqs. (25) at $\lambda = 0.5$, for quantitative comparison with the full field simulation in Sec. 3. We solve numerically the coupled equations (25) for initial conditions matching those in the full simulation of Sec. 3. While iterating these equations in time, the radius $R$ changes and we have to follow the dependence of $q, g_{XX}, g_{RR}$ on $R$ that is shown in Figs. 7, 8. This is most conveniently achieved by using $q(R) = 2/K_1(R)$ as in Fig. 7 and polynomial functions that fit the data for $g_{XX}(R)$, $g_{RR}(R)$.

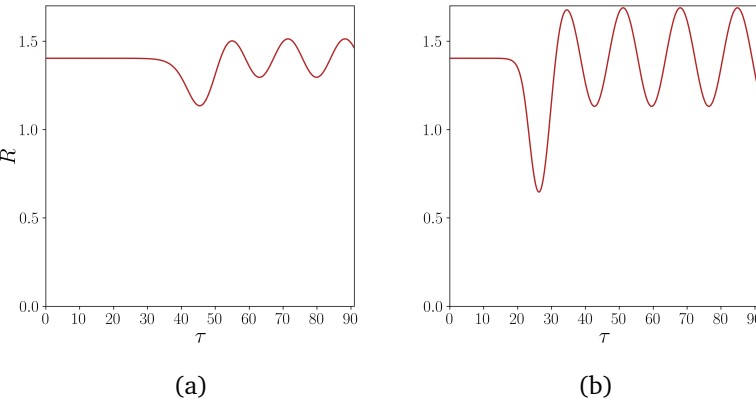

Figure 10: The radius of the skyrmions during the collision according to the collective co-ordinate model for the cases with initial velocities (a) $v = 0.2$ and (b) $v = 0.4$. The same initial conditions for initial velocity and distance of skyrmions are used as in Fig. 5.

The results from the solution of Eqs. (25) are plotted in Figs. 9, 10. They agree with the full simulations in their main features. The skyrmions initially travel with a constant velocity, they decelerate due to repulsion and their radius is reduced at the same time. They eventually bounce back and acquire almost their initial speed but the radius is now oscillating around its equilibrium value, meaning that the breathing mode has been excited. The velocity is constant despite the oscillation of the radius, as is seen in Eq. (25a) since $\partial V_{\mathrm{int}}/\partial X$ decays rapidly with $X$. The period of oscillations matches the one observed in the full simulations for $v = 0.2$, but it does not reproduce this very well for $v = 0.4$. This is attributed to the fact that, for $v = 0.4$, the full simulations give periodic variation of $R$, as seen in Fig. 5, that apparently contain multiple modes beyond the breathing. The discrepancy can be anticipated as the construction of the collective co-ordinate model guarantees its accuracy only for small velocities.

A further possibility arises when the skyrmions have enough kinetic energy to climb up to the value $V_{\mathrm{el}}(R = 0) = 4\pi$. In that case, $R$ may be reduced to zero during the dynamics, meaning that the skyrmions collapse. This does indeed happen as we have seen in the simulations of Section 3. It can be reproduced within the model (25) when a large enough initial velocity ($v \approx 0.622$) is used.

Note that two skyrmions cannot go on top of each other in the full field model, as this would give a singular configuration, and consequently the interaction potential has to diverge at close range. This feature is not reproduced by the interaction potential used in the collective co-ordinate model as it lies outside the region of validity of Eq. (20). All our simulations stay within this range.

## 5 Conclusions

Exploiting the solitary wave properties of skyrmions in antiferromagnets, we have studied head-on collisions between two skyrmions. As a result of the interaction, skyrmions reduce their size and two outcomes are possible. Skyrmions that have an energy $E < 4\pi$ collide and bounce back eventually almost restoring their initial speed. Skyrmions that have an energy $E > 4\pi$ shrink as they approach until they concentrate to a point and disappear. We describe the collision process and explain the main features by following the kinetic energy within the sigma model for antiferromagnets. We have derived a formula for the calculation of the kinetic

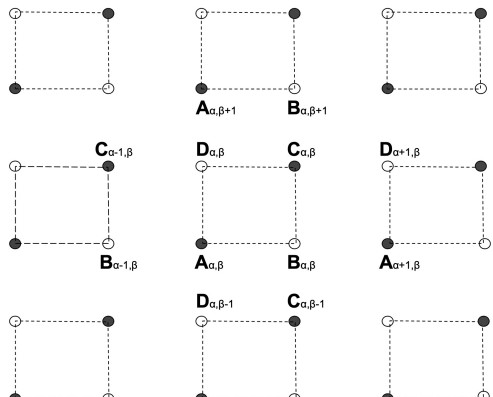

Figure 11: A tetramerization of the square lattice. The tetramers are indexed by integers $\alpha, \beta$ and the spins at each tetramer are denoted by $A, B, C, D$.

energy in terms of the spin vectors in the discrete system. The calculation of the kinetic energy in the lattice is a nontrivial task that is interesting in its own right. We develop a collective co-ordinate model for the distance between the skyrmions and their radius and give a description of the collision process. The model faithfully reproduces the main features of the collision process seen in the field simulations. The elastic property of the skyrmion, that is included in the collective co-ordinate model, emerges as an important factor for the soliton collision.

The dynamical behavior of skyrmions in antiferromagnets is dramatically different than in ferromagnets. The results presented in this paper show how these dynamics couple with interactions between skyrmions to produce behaviors that have not been observed or predicted before in magnetic materials. While observations of skyrmions are abundant in ferromagnets, the observation of antiferromagnetic domains is still challenging. Given the rapid development of experimental techniques for the observation of antiferromagnetic order, we may reasonably expect that the creation and observation of skyrmions will be achievable in the next generation of experiments.

## Acknowledgments

**Funding information** This work was supported by the project "ThunderSKY" funded by the Hellenic Foundation for Research and Innovation and the General Secretariat for Research and Innovation, under Grant No. 871. Bruno Barton-Singer acknowledges funding from the ERC under the European Union's Horizon Europe research and innovation program (Grant No. 101078061, Project "SINGinGR").

## A Continuum energy from the energy of the spin lattice

We can obtain the continuous form of the energy (9) from the discrete form (1). The tetramerization of the lattice is shown in Fig. 11.

For the exchange energy in the discrete, it is sufficient to consider the interactions between the spins in each tetramer. There are four terms and they can be compactly written as

$$(\mathbf{A} + \mathbf{C}) \cdot (\mathbf{B} + \mathbf{D}) = 4s^2 (\boldsymbol{m} + \boldsymbol{n}) \cdot (\boldsymbol{m} - \boldsymbol{n}) = 4s^2 (\boldsymbol{m}^2 - \boldsymbol{n}^2) = 4s^2 (2\boldsymbol{m}^2 + \mathbf{k}^2 + \mathbf{l}^2 - 1).$$

In order to take into account also the interaction with spins in neighboring tetramers, we have to double the above result. For small $\epsilon$, we can obtain a continuum approximation by setting

$$\sum_{\alpha,\beta} \to \frac{1}{(2\epsilon)^2} \int d^2x\,. \tag{A.1}$$

The exchange energy in the discrete is then approximated as

$$E_{\text{ex}}^d \approx 8Js^2 \frac{1}{(2\epsilon)^2} \int (2\boldsymbol{m}^2 + \mathbf{k}^2 + \mathbf{l}^2)\, d^2x = Js^2 \frac{1}{2} \int (\dot{\boldsymbol{n}}^2 + \boldsymbol{n}_x^2 + \boldsymbol{n}_y^2)\, d^2x = Js^2 (E_{\text{kin}} + E_{\text{ex}})\,, \tag{A.2}$$

where we omitted a constant and used Eq. (8). Eq. (A.2) gives the decomposition of the exchange energy in the discrete to the kinetic and exchange energy in the continuum. A more elaborate approach where the interaction of spins of one tetramer are involved in the calculation explicitly, gives the same result (A.2), and we will thus not give it in detail here.

We may now derive a formula for the kinetic energy so that it can be calculated over the numerical mesh used in simulations. Using Eq. (8), we have

$$\boldsymbol{m} = \frac{\epsilon}{2\sqrt{2}} \boldsymbol{n} \times \dot{\boldsymbol{n}} \Rightarrow \boldsymbol{m}^2 = \frac{\epsilon^2}{8} \dot{\boldsymbol{n}}^2\,,$$

that is substituted in Eq. (9) for the kinetic energy of the continuum model, to give

$$E_{\text{kin}} = \frac{4}{\epsilon^2} \int \boldsymbol{m}^2\, d^2x\,. \tag{A.3}$$

The form for calculations on the numerical mesh is obtained by applying the substitution (A.1),

$$E_{\text{kin}} \approx 16 \sum_{\alpha,\beta} (\boldsymbol{m}_{\alpha,\beta})^2\,. \tag{A.4}$$

A second, equivalent, formula can be derived for the kinetic energy. Eq. (8) for $\boldsymbol{m}$ is equivalent to

$$\epsilon \dot{\boldsymbol{n}} = 2\sqrt{2}\, \boldsymbol{n} \times \boldsymbol{m} \Rightarrow \epsilon \dot{\boldsymbol{n}} = \frac{1}{2\sqrt{2}} \left[ (\mathbf{A} + \mathbf{C}) \times (\mathbf{B} + \mathbf{D}) \right]\,, \tag{A.5}$$

where we have used the definitions of $\boldsymbol{m}, \boldsymbol{n}$ in Eq. (4). A form for the calculation of the kinetic energy (A.4) on the numerical mesh is obtained by applying (A.1),

$$E_{\text{kin}} = \frac{1}{2} \int \dot{\boldsymbol{n}}^2\, d^2x \approx \frac{1}{4} \sum_{\alpha,\beta} \left[ (\mathbf{A}_{\alpha,\beta} + \mathbf{C}_{\alpha,\beta}) \times (\mathbf{B}_{\alpha,\beta} + \mathbf{D}_{\alpha,\beta}) \right]^2\,. \tag{A.6}$$

For completeness, we will give the relation between the discrete and continuum energy for the DM and anisotropy terms. For the DM energy, we write

$$\begin{aligned}
E_{\text{DM}}^d &= D \sum_{\alpha,\beta} [\hat{\boldsymbol{e}}_2 \cdot [\mathbf{B}_{\alpha,\beta} \times (\mathbf{A}_{\alpha+1,\beta} - \mathbf{A}_{\alpha,\beta}) + \mathbf{C}_{\alpha,\beta} \times (\mathbf{D}_{\alpha+1,\beta} - \mathbf{D}_{\alpha,\beta})] \\
&\quad - \hat{\boldsymbol{e}}_1 \cdot [\mathbf{D}_{\alpha,\beta} \times (\mathbf{A}_{\alpha,\beta+1} - \mathbf{A}_{\alpha,\beta}) + \mathbf{C}_{\alpha,\beta} \times (\mathbf{B}_{\alpha,\beta+1} - \mathbf{B}_{\alpha,\beta})] \\
&\approx 2Ds^2 \sum_{\alpha,\beta} \hat{\boldsymbol{e}}_1 \cdot (\boldsymbol{n}_{\alpha,\beta} \times \boldsymbol{n}_{\alpha+1,\beta}) - \hat{\boldsymbol{e}}_2 \cdot (\boldsymbol{n}_{\alpha,\beta} \times \boldsymbol{n}_{\alpha+1,\beta})\,,
\end{aligned}$$

where we have retained only the lowest order terms in the last equation. Applying Eq. (A.1), we obtain

$$E_{\text{DM}}^d \approx Js^2 E_{\text{DM}}\,. \tag{A.7}$$

For the anisotropy energy, we write

$$
\begin{aligned}
E_{\mathrm{a}}^{d} &= \frac{K}{2} \sum_{\alpha,\beta} \left( A_{\alpha,\beta,3}^2 + B_{\alpha,\beta,3}^2 + C_{\alpha,\beta,3}^2 + D_{\alpha,\beta,3}^2 \right) \\
&= 2Ks^2 \sum_{\alpha,\beta} \left( n_{\alpha,\beta,3}^2 + m_{\alpha,\beta,3}^2 + k_{\alpha,\beta,3}^2 + l_{\alpha,\beta,3}^2 \right) \approx 2Ks^2 \sum_{\alpha,\beta} n_{\alpha,\beta,3}^2 \,,
\end{aligned}
$$

where subscript 3 denotes the third component of each vector, and we have retained only the lowest-order terms in the last equation. The continuum approximation is

$$
E_{\mathrm{a}}^{d} \approx \frac{Ks^2}{\epsilon^2} \left[ \frac{1}{2} \int (n_3)^2 d^2x \right] = Js^2 \, E_{\mathrm{a}} \,. \tag{A.8}
$$

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
