# Peer review of "Interaction and collision of skyrmions in chiral antiferromagnets"

_SciPost Physics, doi:SciPost Phys. 18, 037 (2025)_

## Round 1 · Referee Report · Anonymous (Referee 1) · 2024-4-29

Strengths

1-clearly written and structured
2-numerical discovery of new type of skyrmion scattering with associated scale change in antiferromagnetic materials
3- clear discussion of the results, with good set of references

Weaknesses

1-lack of an understanding of the scale change during the skyrmion scattering

Report

This is an interesting account of a novel phenomenon in magnetic skyrmion interactions. While this adds an interesting phenomenon to the long list of different types of skyrmion interaction behaviour, it is hardly a groundbreaking discovery. Neither does it open new synergetic links or pathways for skyrmion research. In that sense it does not meet the expectations of SciPost

Requested changes

Towards the end of the paper the authors begin a discussion of how one might understand the interaction behaviour they observe numerically. This discussion is merely a sketch at the moment: it contains unjustified assumptions (`Assuming that f_2 is larger ...'). The authors should justify this assumption and pursue the analytical study to produce an understanding of the interaction behaviour.

Recommendation

Ask for major revision

---

## Round 1 · Referee Report · Elizaveta Tremsina (Referee 2) · 2024-5-23

Strengths

1. Work is well presented
2. Model and method clearly presented
3. Interesting collision cases presented

Weaknesses

1. Missing key references to past work in introduction, especially regarding chiral skyrmion dynamics.
10.1103/PhysRevLett.116.147203
10.1103/PhysRevB.106.L220402
10.1126/SCIENCE.ADD5751

2. No mention of skyrmion mass or inertia, perhaps this is relevant point to consider when talking about energetics of collisions.
10.1103/PhysRevB.106.144405

3. The effective field approach provided in section 3 seems like a nice simple model but it is unclear how it is actually used in the simulations to make sense of the collisions or predict their various outcomes. Thus, it is difficult to objectively judge the validity and usefulness of that portion of the work.

Additionally, perhaps it would be useful to tie in past work on skyrmion-skyrmion interactions, in which the potential function was derived analytically, and see how it relates to the field analytics of section 3.
10.1038/s42005-023-01145-9

Report

The paper gives a nice overview of the motivation and simulation method used, as well as describing the observed collisions and their aftermath. However, it is missing information on the larger implications of this work, specifically, for use in computing applications and linking with larger body of work done with skyrmions in magnetic materials. Lastly, the paper seems to ignore some relevant work done on dynamics of chiral skyrmions.

Requested changes

1. Add references to past work
2. Explain the exact choice of material parameters used
3. Address the weakness points listed above

Recommendation

Ask for major revision

---

## Round 2 · Referee Report · Anonymous (Referee 1) · 2024-11-15

Strengths
2-Good combination of numerical work with intuitive understanding, based on a simple collective coordinate model
3-Clear structure and exposition
Weaknesses
2-Could have explored wider range of scattering processes (eg varying impact parameter)
Report
Requested changes
1-Clarify the form of `skyrmion' in this model i.e., the precise form of n in (13) and (10)
2-Briefly comment if the breathing of travelling skyrmions leads to oscillations in the linear velocities. The dependence of the mass on the scale R would imply this, but it is not visible in the plots in Figure 9
Recommendation
Publish (meets expectations and criteria for this Journal)

---

## Round 2 · Author Response

In the resubmission, the manuscript has been completed extensively in a very significant way, where a colleague (who is now a co-author) has made the central contributions. We have added a Section (4) for the understanding of skyrmion scattering via a collective co-ordinate approach. This shows the origin of the scale change of skyrmions during scattering and the role that it plays in the dynamics. Virtually all significant features of the collision dynamics are now discussed in detail. The collective co-ordinate dynamics faithfully reproduces the full-field (simulation) results.
The theoretical approach involves results from soliton theory and from field theories outside magnetism or condensed matter. This is reflected in the list of references (e.g., in Section 4). It demonstrates in a magnetic system (which is realistic and has potential for applications) a type of skyrmion interaction that changes the current point of view in the wider soliton community, as it shows the substantial role of a very specific internal skyrmion mode (breathing mode) for the skyrmion interaction and scattering. We therefore believe that it will open a pathway for follow-up works including synergetic ones.
Some references that were very relevant to this work were unfortunately missing in the previous manuscript and they now had to be added. On the other hand, works on skyrmions in synthetic antiferromagnets (SAF) are not cited in the present paper. We think that SAF are not closely enough related to standard AFM especially when dynamics are studied (a common theoretical model cannot be obtained as far as we know). SAF skyrmions are called AFM skyrmions in some works, but we prefer to avoid citations of SAF works, which means that we believe that these two systems (each one very interesting in its own right) can only be studied separately.
We hope that the manuscript, in its present completed form, will be found to meet the criteria for publication in SciPost Physics.

---

## Round 2 · List of Changes

In response to the remarks of Report 1
Section 4 has been added which gives an analytical study that produces an understanding of the interaction behaviour. The skyrmion scale change during the scattering is a central part of the analytics. In Sec. 3, at the end, a paragraph that was heuristically describing the interaction behaviour has been mostly erased.
In response to the remarks of Report 2
References have been added (but not the ones studying SAF skyrmions, as explained in our response letter).
Skyrmion mass is now discussed in Section 4 (e.g., in connection to Fig. 8a).
Section 4 now provides a much more precise discussion for skyrmion interaction.
All main changes
Bruno Barton-Singer was added as co-author.
The abstract was completed to include the analytical results.
Sec. 1. References have been added (now [23], [26], [30]).
Sec. 3. The last paragraph (on heuristic description of skyrmion interaction) has been mostly erased.
Section 4 has been added. It includes Figs. 7,8,9,10 and a number of new References.
Sec. 5 (Conclusions). A phrase has been added at the end of the first paragraph.

---

## Editorial Decision

published